# MC180295 Inhibited Epstein–Barr Virus-Associated Gastric Carcinoma Cell Growth by Suppressing DNA Repair and the Cell Cycle

**DOI:** 10.3390/ijms231810597

**Published:** 2022-09-13

**Authors:** Tomohiro Fujii, Jun Nishikawa, Soichiro Fukuda, Naoto Kubota, Junzo Nojima, Koichi Fujisawa, Ryo Ogawa, Atsushi Goto, Koichi Hamabe, Shinichi Hashimoto, Aung Phyo Wai, Hisashi Iizasa, Hironori Yoshiyama, Kohei Sakai, Yutaka Suehiro, Takahiro Yamasaki, Taro Takami

**Affiliations:** 1Faculty of Laboratory Science, Yamaguchi University Graduate School of Medicine, Ube 755-8505, Japan; 2Department of Gastroenterology and Hepatology, Yamaguchi University Graduate School of Medicine, Ube 755-8505, Japan; 3Department of Microbiology, Shimane University Faculty of Medicine, Izumo 693-8501, Japan; 4Department of Oncology and Laboratory Medicine, Yamaguchi University Graduate School of Medicine, Ube 755-8505, Japan

**Keywords:** gastric cancer, Epstein–Barr virus, DNA methylation, MC180295, demethylating agent

## Abstract

DNA methylation of both viral and host DNA is one of the major mechanisms involved in the development of Epstein–Barr virus-associated gastric carcinoma (EBVaGC); thus, epigenetic treatment using demethylating agents would seem to be promising. We have verified the effect of MC180295, which was discovered by screening for demethylating agents. MC180295 inhibited cell growth of the EBVaGC cell lines YCCEL1 and SNU719 in a dose-dependent manner. In a cell cycle analysis, growth arrest and apoptosis were observed in both YCCEL1 and SNU719 cells treated with MC180295. MKN28 cells infected with EBV were sensitive to MC180295 and showed more significant inhibition of cell growth compared to controls without EBV infection. Serial analysis of gene expression analysis showed the expression of genes belonging to the role of *BRCA1* in DNA damage response and cell cycle control chromosomal replication to be significantly reduced after MC180295 treatment. We confirmed with quantitative PCR that the expression levels of *BRCA2, FANCM, RAD51, TOP2A*, and *CDC45* were significantly decreased by MC180295. *LMP1* and *BZLF1* are EBV genes with expression that is epigenetically regulated, and MC180295 could up-regulate their expression. In conclusion, MC180295 inhibited the growth of EBVaGC cells by suppressing DNA repair and the cell cycle.

## 1. Introduction

Epstein–Barr virus (EBV) is a ubiquitous human herpes virus closely associated with both lymphoid and epithelial malignancies, and its association with nasopharyngeal carcinoma and gastric carcinoma is well-established [1]. Gastric cancer was responsible for over one million new cases of cancer in 2020 and an estimated 769,000 deaths, ranking fifth for incidence and fourth for mortality globally [2]. About 9% of gastric cancers have been identified as EBV-positive. Thus, EBV-associated gastric carcinoma (EBVaGC) is the most common cancer among EBV-related malignancies [3].

The prognosis for unresectable recurrent gastric cancer is extremely poor. Chemotherapy for gastric cancer is comprised mainly of cytotoxic agents, and molecularly targeted agents are still rare. Only programmed cell death 1 (*PD-1*) and human epidermal growth factor receptor 2 have recently become new targets in advanced gastric cancer [4]. In addition, the tumor microenvironment has emerged as a potential therapeutic target in several malignancies including gastric cancer [5]. Several agents targeting the tumor microenvironment are currently under assessment in both preclinical and clinical studies. The development of chemotherapy based on the pathogenesis of gastric cancer is clearly desired.

The Cancer Genome Atlas (TCGA) divides gastric adenocarcinomas into four molecular subtypes: (1) EBVaGC, (2) microsatellite instability (MSI), (3) chromosomal instability, and (4) genomically stable tumors. The characteristics of EBVaGC are reported to include the harboring of recurrent *PIK3CA* mutations, DNA hypermethylation, amplification of *JAK2*, and overexpression of *PD-L1* and *PD-L2* [6]. Viral DNA methylation regulates EBV latency type and inhibits the expression of EBV latent genes that are possible targets of cytotoxic T lymphocytes [7,8]. Methylation of host cell DNA inactivates tumor suppressor genes and tumor-associated antigens. Methylation of both viral and host DNA is one of the major mechanisms involved in the development of EBVaGC [9,10].

MC180295 is an aminothiazole compound that was discovered by screening for demethylating agents. Zhang et al. reported that MC180295 acted as a specific inhibitor of cyclin dependent kinase 9 (*CDK9*) and reactivated epigenetically silenced genes in multiple cancer cells [11]. We verified the effect of this new *CDK9* inhibitor, MC180295, on EBVaGC in this study.

## 2. Results

### 2.1. Growth Inhibition by MC180295 in EBV-Associated Gastric Cancer Cells

MC180295 inhibited cell growth of the EBVaGC cell lines YCCEL1 and SNU719 in a dose-dependent manner. The inhibition of cell growth of YCCEL1 by MC180295 was significant at 4 days after exposure to 1 μM MC180295 (Figure 1a). The inhibition of cell growth of SNU719 by MC180295 was significant at 2 to 4 days after exposure to 1 μM MC180295 (Figure 1b). MKN28 cells infected with EBV were sensitive to MC180295 and showed more significant inhibition of cell growth compared to EBV-uninfected MKN28 cells and controls treated with DMSO (Figure 1c).

In a cell cycle analysis, a concomitant decrease in the proportion of cells in the S phase was observed in both SNU719 and YCCEL1 cells treated with MC180295: from 15.6% to 5.4% in SNU719 and from 31.0% to 9.1% in YCCEL1. The sub-G1 fraction increased from 0.9% to 12.6% in SNU719 cells and from 1.5% to 21.0% in YCCEL1 at 48 h after MC180295 treatment. In YCCEL1 cells, the percentage of cells in the sub-G1 fraction increased to 26.4% at 72 h after MC180295 treatment (Figure 2).

### 2.2. Patterns of Gene Expression after MC180295 Treatment

The results of serial analysis of gene expression (SAGE) were integrated by Ingenuity Pathways Analysis. We found two pathways in which gene expression varied significantly after MC180295 treatment. The expressions of genes belonging to the pathways “role of *BRCA1* in DNA damage response” and “cell cycle control chromosomal replication” were significantly reduced in both YCCEL1 and SNU719 cells. Table 1 shows the change of gene expression of the pathway “role of *BRCA1* in DNA damage response”. We selected *FANCM*, *BRCA2*, and *RAD51* for real-time PCR. Table 2 shows the change of gene expression of the pathway “cell cycle control of chromosomal replication”. The genes down-regulated by MC180295 were common in both YCCEL1 and SNU719 cells. We selected *TOP2A* and *CDC45* for real-time PCR.

In quantitative PCR, the expression levels of *BRCA2*, *FANCM*, and *RAD51* as well as those of *TOP2A* and *CDC45*, decreased in both the YCCEL1 and SNU719 cells treated with MC180295 (Figure 3).

### 2.3. Protein Expression of γH2A Histone Family Member X(γH2AX) and Caspase-3 after MC180295 Treatment

Expression of proteins involved in DNA repair and cell cycle regulation were investigated. In SNU719 cells treated with MC180295, expression of γH2A histone family member X (γH2AX) showed an increase up to 24 h but showed a slight decrease at 48 h. In YCCEL1 cells, MC180295 treatment increased γH2AX expression up to 48 h. In SNU719 cells treated with MC180295, expression of cleaved caspase-3 showed an increase up to 24 h but showed a slight decrease at 48 h. In YCCEL1 cells, MC180295 treatment increased cleaved caspase-3 expression at 48 h (Figure 4).

### 2.4. Expression of EBV Genes LMP1 and BZLF1 after MC180295 Treatment

The mRNA expression levels of *LMP1* and *BZLF1* were increased by 1 μM MC180295 in both the YCCEL1 and SNU719 cells (Figure 5).

## 3. Discussion

*Helicobacter pylori* and EBV are recognized as risk factors for gastric cancers. Infection by both induces epigenetic alterations in gastric mucosal cells and promotes the development of gastric cancer [12]. Thus, epigenetic treatment using demethylating agents would seem to be promising. There are few studies on clinical trials of demethylating agents for gastric cancers. A Phase I trial using 5-azacitidine prior to neoadjuvant chemotherapy was reported in 2017 and showed the overall response rate to be 67%, with 25% of patients achieving complete response [13]. A Phase II trial is scheduled, which uses a modified regimen. Another phase II clinical trial combining histone deacetylase (HDAC) inhibitor SAHA could not show improvement in the clinical outcome of the patients with metastatic or unresectable gastric cancer [14]. Thus, epigenetic strategies for gastric cancer treatment have not been fully achieved.

MC180295 is an aminothiazole compound found by screening for demethylating agents [11]. MC180295 can reactivate epigenetically silenced genes by remodeling chromatin but without affecting DNA methylation, as with HDAC inhibitors. The expression pattern of the genes induced by MC180295 was broadly similar to what was seen with DNA methyltransferase (DNMT) inhibition [11]. Through TCGA molecular classification, both EBVaGC and the MSI subtype were identified as CpG island methylator phenotype (CIMP) [6]. Thus, MC180295 could be a novel candidate for treatment of gastric cancer with CIMP. We showed growth inhibition of EBVaGC cell lines by MC180295 treatment. Some cells treated with MC180295 were arrested, and apoptosis was induced. To our knowledge, this study is the first report to reveal the effectiveness of MC180295 in inhibiting tumor cell proliferation in EBVaGC. A clinical study is underway to evaluate its safety. If MC180295 can be administered to patients with EBVaGC, it may restore expression of tumor suppressor genes with expression that is suppressed by DNA methylation and histone modification [9]. Tumor-associated antigens are also down-regulated in EBVaGC through epigenetic mechanisms [10], and restoration of their expression would be an attractive approach for possible anti-tumor immunity or for combination therapy with immune checkpoint inhibitors. We believe that this would be a major advance in the chemotherapy of unresectable EBVaGC.

We analyzed patterns of gene expression after MC180295 treatment in EBVaGC. Common pathways were significantly suppressed in both the SNU719 and YCCEL1 cell lines. *BRCA2* and *RAD51*, which cooperate with *BRCA1* for homologous recombination [15], were significantly down-regulated by MC180295. As *BRCA1* recruitment to damaged DNA sites is dependent on *CDK9* [16], MC180295 might interfere with the repair of double-strand DNA breaks by acting as a *CDK9* inhibitor.

*BRCA1*- and *BRCA2*-deficient cells are sensitive to poly (ADP-ribose) polymerase (PARP) inhibitors [17]. PARP is a key regulator in the base excision repair process and participates in the repair of single-strand DNA breaks. Loss of PARP activity is likely to cause the accumulation of single-strand breaks, which are converted to double-strand DNA breaks during replication or homologous recombination repair [18,19]. The increase in damaged DNA results in the lethality of *BRCA1*- or *BRCA2*-deficient cells. Since MC180295 could suppress the expression of *BRCA1* and *BRCA2*, DNA damage response for double-stranded DNA breaks was disrupted in the MC180295-treated EBVaGC cells, indicating that the combination of MC190285 and a PARP inhibitor might lead to cancer-cell-specific death.

The genes associated with DNA replication, including *TOP2A* and *CDC45*, were down-regulated after MC180295. *TOP2A* loosens double-stranded DNA by changing the helical structure in processes of DNA replication and transcription [20]. *CDC45* plays a central role in the regulation of the initiation and elongation stages of chromosomal DNA replication [21]. Knockdown of *TOP2A* and *CDC45* has been shown to suppress the growth of various cancer cells [22,23]. Cell cycle analysis showed a decrease in the S phase and G1 arrest after MC180295 treatment. These results are thought to be due to the inhibition of DNA replication.

Expression of proteins involved in DNA repair and cell cycle regulation were investigated. γH2AX is a sensitive marker of DNA damage response [24,25]. Caspase-3 plays important role in the process of apoptosis, and detection of cleaved caspase-3 indicates the occurrence of apoptosis [25]. The expression of cleaved caspase-3 and γH2AX increased in a similar fashion after MC180295 treatment. The result showed that the data from the cell cycle analysis were convincing.

Induction of the lytic cycle is one of the therapeutic targets for EBV-related malignancies. *BZLF1* is a transcriptional activator that binds to AP-1-like motifs in the promoters of early lytic genes and induces lytic infection in latently EBV-infected cells [26]. *BZLF1* is epigenetically regulated, and demethylating agents and HDAC inhibitors are known to induce *BZLF1* expression in EBV-carrying cells [27]. *LMP1* is also a viral gene with expression that is epigenetically regulated [28]. *BZLF1* has been shown to be a binding partner of several DNA damage response proteins, such as *53BP1*, *Ku80*, *p53*, and *RNF8* [29,30,31,32,33]. Moreover, screening of the EBV proteins that inhibit DNA damage response identified EBV *BKRF4* as a DNA damage response inhibitor. Since *BKRF4* was expressed in both latent and lytic EBV infections, several EBV genes expressed during both lytic and latent infection might be involved with DNA damage response [34].

The limitation of this study is that we only examined the growth-inhibitory effect of MC180295 in EBVaGC lines in vitro. In the future, we are currently planning to investigate the effect of MC180295 on cancer cell lines transplanted into immunodeficient mice.

## 4. Materials and Methods

### 4.1. Cell Culture

The EBV-associated gastric cancer cell lines YCCEL1 and SNU719 were cultured in medium with 10% FBS. YCCEL1 cells were seeded at a density of 5 × 10^3^ cells/mL onto 96-well dishes, and SNU719 cells were seeded at a density of 1 × 10^4^ cells/mL onto 96-well dishes. Cells were treated with MC180295 (Selleck Chemicals, Houston, TX, USA) or DMSO as a control. MC180295 was diluted to medium at concentrations of 10 μM, 1 μM, 0.5 μM, and 0.1 μM. Cell viability was analyzed by MTS (3-(4,5-dimethylthiazol-2-yl)-5-(3-carboxymethoxyphenyl)-2-(4-sulfophenyl)-2H-tetrazolium) assay. MTS was added to the well, and the absorbance was measured at 490 nm. Data were obtained from three independent experiments.

We previously infected EBV recombinants that carried NeoR to the EBV-negative gastric carcinoma cell line MKN28 [35]. MKN28 cells infected with EBV and uninfected control cells were treated with 1 μM MC180295 for 48 h. The harvested cells were then seeded into 96-well plates and cultured in medium containing 1 μM MC180295, and cell proliferation was assessed by MTS.

### 4.2. Cell Cycle Analysis

Cell cycle analysis was performed for cells treated in the presence or absence of 1 μM MC180295 for 48 h or 72 h. For measurement of DNA content, treated cells were collected and fixed with 70% ethanol. After resuspension, fixed cells were washed and incubated with propidium iodide and RNase. After staining, flow cytometry was used to analyze the cell cycle phase.

### 4.3. RNA Extraction and SAGE Analysis

The total RNA of cells treated in the presence or absence of 1 μM MC180295 for 48 h was isolated using an All Prep DNA/RNA Mini Kit (Qiagen, Hilden, Germany), in accordance with the protocol of the manufacturer. For SAGE, we cultured YCCEL1 and SNU719 cells and treated them with 1 μM MC180295. After incubation for 48 h, mRNA was extracted from the cells. We synthesized cDNA from mRNA using reverse transcriptase. Gene expression was then comprehensively investigated from the obtained cDNA by SAGE. Based on the results of Ingenuity Pathways Analysis by SAGE, the expression levels of the genes that changed significantly were individually quantified by real-time PCR.

### 4.4. Real-Time PCR

Real-time RT-PCR was performed with Applied Biosystems SYBR Green I using a LightCycler (Roche Diagnostics, Indianapolis, IN, USA) to quantify the mRNA. The primer sequences of FANCM, BRCA2, RAD51, TOP2A, CDC45, LMP1, BZLF1, and β-actin are listed in Table 3. Data were calculated as a ratio in reference to the mRNA of β-actin. Data were obtained from three independent experiments.

### 4.5. Western Blot for γH2AX and Caspase-3

SNU719 or YCCEL1 cells were treated with 1 μM of MC180295 and lysed by RIPA buffer (Thermo Fisher Scientific, Waltham, MA, USA), supplemented with protease inhibitor (cOmplete mini, Sigma-Aldrich, St. Louis, MO, USA) and phosphatase inhibitor (PhosStop, Sigma, St. Louis, MO, USA). Five μg of protein samples were electrophoresed on 15% SDS-page gel and transferred to PVDF membranes (Millipore, Billerica, MA, USA). The membranes were blocked by BlockAce or Tris buffered saline with 0.01% tween 20 and 5% bovine serum albumin. Then, the membranes were incubated with antibodies specific to anti-phospho-histone H2A.X (Ser139) (γH2AX) (#2577; Cell Signaling Technology, Danvers, MA, USA), anti-caspase-3 (W20054B; Biolegend, San Diego, CA, USA), anti-cleaved caspase-3 (Asp175) (5A1E; Cell Signaling, Danvers, MA, USA), and anti-β-actin (AC-15, Sigma, St. Louis, MO, USA). After washing, the membranes were incubated with horseradish peroxidase (HRP)-linked anti-rabbit IgG (Cell Signaling, Danvers, MA, USA), HRP-linked anti-rat IgG (Beckman Coulter, Brea, CA, USA), and HRP-linked anti-mouse IgG (Cell Signaling, Danvers, MA, USA). Specific bands were visualized using an Immobilon (Millipore, Billerica, MA, USA) and were detected by X-ray film. Anti-β-actin antibody was used as an internal control.

### 4.6. Statistical Analysis

Results were analyzed statistically by the Student’s t-test, and a value of *p* < 0.05 was considered to indicate statistical significance. Analysis was performed using StatFlex version 6.0 statistical software (Artech Co., Ltd., Osaka, Japan).

## 5. Conclusions

MC180295 inhibited the growth of EBVaGC cells by suppressing DNA repair and the cell cycle.

## Figures and Tables

**Figure 1 ijms-23-10597-f001:**
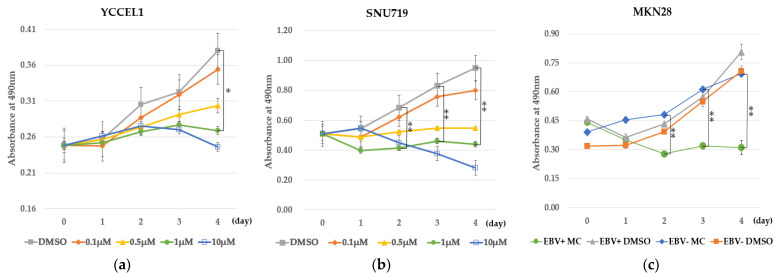
MC180295 inhibits cell growth in EBV-positive gastric cancer cells. (**a**) The inhibition of cell growth of YCCEL1 by MC180295 was significant at 4 days after exposure to 1 μM MC180295. (**b**) The inhibition of cell growth of SNU719 by MC180295 was significant at 2 to 4 days after exposure to 1 μM MC180295. (**c**) EBV-positive MKN28 cells showed marked growth inhibition by 1 μM MC180295, compared to EBV-uninfected MKN28 cells and controls treated with DMSO. The y-axis represents absorbance at 490 nm in the MTS assay. Data were obtained from three independent experiments. Bars denote SE (*n* = 3), * *p* < 0.05. ** *p* < 0.01.

**Figure 2 ijms-23-10597-f002:**
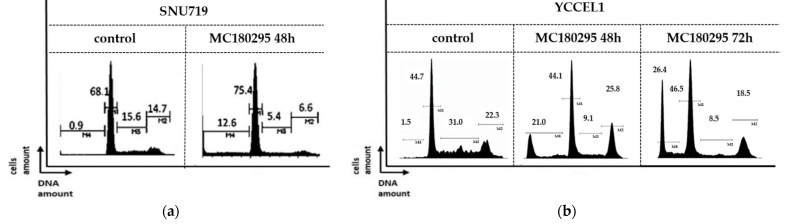
Cell cycle analysis after MC180295 treatment in SNU719 (**a**) and YCCEL1 (**b**). The percentage of cells in S phase decreased, and those in the sub-G1 fraction increased at 48 h after treatment. In YCCEL1 cells, the percentage of cells in the sub-G1 fraction increased to 26.4% at 72 h after MC180295 treatment.

**Figure 3 ijms-23-10597-f003:**
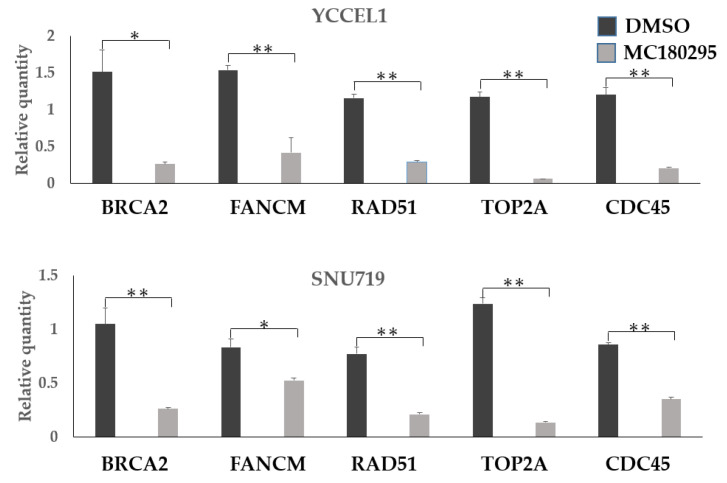
Gene expression levels after MC180295 treatment. The expression levels of *BRCA2*, *FANCM*, and *RAD51* decreased in both YCCEL1 and SNU719 cells treated with MC180295, as well as those of *TOP2A* and *CDC45*. Data were obtained from three independent experiments. Bars denote SE (*n* = 3), * *p* < 0.05. ** *p* < 0.01.

**Figure 4 ijms-23-10597-f004:**
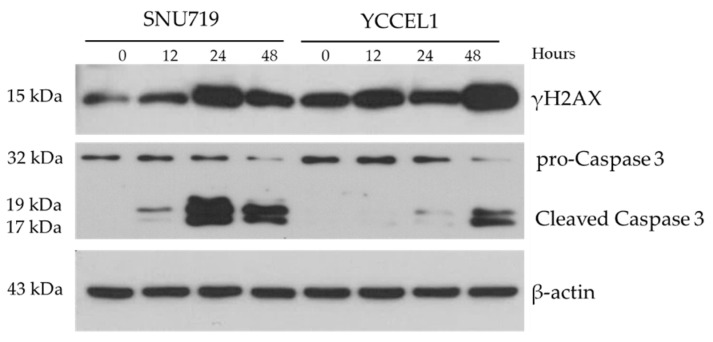
Protein Expression of γH2A histone family member X (γH2AX) and caspase-3 after MC180295 treatment. The expression levels of γH2AX and cleaved caspase-3 were up-regulated in both YCCEL1 and SNU719 cells treated with MC180295.

**Figure 5 ijms-23-10597-f005:**
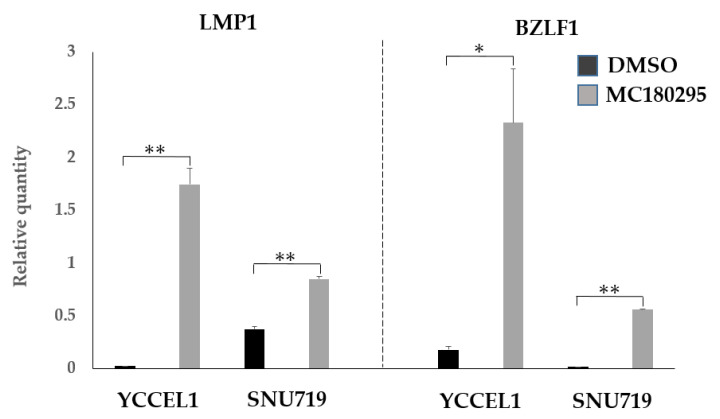
EBV gene expression after MC180295 treatment. The expression levels of *LMP1* and *BZLF1* were up-regulated in both YCCEL1 and SNU719 cells treated with MC180295. Data were obtained from three independent experiments. Bars denote SE (*n* = 3), * *p* < 0.05. ** *p* < 0.01.

**Table 1 ijms-23-10597-t001:** Change of gene expression of pathway “role of BRCA1 in DNA damage response”.

		Expression Fold Change	
Symbol	Entrez Gene Name	YCCEL1	SNU719
E2F8	E2F transcription factor 8	−17.625	−59.15
FANCM	FA complementation group M	−13.337	−12.558
BRCA2	BRCA2 DNA repair associated	−10.78	−24.684
BLM	BLM RecQ-like helicase	−6.618	−10.5
RAD51	RAD51 recombinase	−6.016	−9.869
PLK1	Polo-like kinase 1	−5.603	−4.065
FANCA	FA complementation group A	−5.155	−5.353
E2F1	E2F transcription factor 1	−5.017	−6.087
MSH6	mutS homolog 6	−4.992	−3.919
BRIP1	BRCA1-interacting protein C-terminal helicase 1	−4.92	−5.305

**Table 2 ijms-23-10597-t002:** Change of gene expression of pathway “cell cycle control of chromosomal replication”.

		Expression Fold Change	
Symbol	Entrez Gene Name	YCCEL1	SNU719
TOP2A	DNA topoisomerase II alpha	−23.748	−72.009
CDC45	Cell division cycle 45	−20.479	−16.356
CDC6	Cell division cycle 6	−6.911	−3.132
MCM6	Minichromosome maintenance complex component 6	−6.511	−5.254
CDC7	Cell division cycle 7	−6.142	−2.967
PRIM1	DNA primase subunit 1	−5.285	−3.952
MCM4	Minichromosome maintenance complex component 4	−4.906	−6.035
POLA2	DNA polymerase alpha 2, accessory subunit	−4.752	−6.194
POLE	DNA polymerase epsilon, catalytic subunit	−4.632	−3.361
MCM9	Minichromosome maintenance 9 homologous recombination repair factor	−4.153	−3.712

**Table 3 ijms-23-10597-t003:** Primers for quantitative real-time PCR.

Genes	Primer Sequence (5′–3′)		Annealing Temperature, °C
BRCA2	Forward	GGCTTCAAAAAGCACTCCAGATG	60
	Reverse	GGATTCTGTATCTCTTGACGTTCC	
FANCM	Forward	GCCGTAAACGTCAAGGCAGGAT	60
	Reverse	CCATCAGGAACCATTCGTGGAC	
RAD51	Forward	TCTCTGGCAGTGATGTCCTGGA	60
	Reverse	TAAAGGGCGGTGGCACTGTCTA	
TOP2A	Forward	GTGGCAAGGATTCTGCTAGTCC	60
	Reverse	ACCATTCAGGCTCAACACGCTG	
CDC45	Forward	TGGATGCTGTCCAAGGACCTGA	60
	Reverse	CAGGACACCAACATCAGTCACG	
LMP1	Forward	AATCTGGATGTATTACCATGGACAAC	60
	Reverse	GCGGGAGGGAGTCATCGT	
BZLF1	Forward	CTGCGCCTCCTGTTGAAG	60
	Reverse	TTAAGAGATCCTCGTGTAAAACATCT	
β-actin	Forward	GCTCCTCCTGAGCGCAAG	60
	Reverse	CATCTGCTGGAAGGTGGACA	

## Data Availability

Not applicable.

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
