# Peer review of "MC180295 Inhibited Epstein–Barr Virus-Associated Gastric Carcinoma Cell Growth by Suppressing DNA Repair and the Cell Cycle"

_ijms, 2022, doi:10.3390/ijms231810597_

Round 1

Reviewer 1 Report

We read with interest this study regarding gastric cancer.

We would suggest the following changes:

  • Introduction section: although the authors correctly included important papers in this setting, we believe a couple of studies should be cited within the introduction (PMID: 33916915 ; PMID: 33916206 ), only for a matter of consistency. We think it might be useful to introduce the topic of this interesting study and to add more data to the background, which appears a bit incomplete.
  • English should be improved and the support of a professional service is suggested.

  • Discussion section: Very interesting and timely discussion. Of note, the authors should expand the Discussion section, including a more personal perspective to reflect on. For example, they could answer the following questions – in order to facilitate the understanding of this complex topic to readers: what potential does this study hold? What are the knowledge gaps and how do researchers tackle them? How do you see this area unfolding in the next 5 years? We think it would be extremely interesting for the readers.

However, we think the authors should be acknowledged for their work. In fact, they correctly addressed an important topic, the methods sound good and their discussion is well balanced.

One additional little flaw: the authors could better explain the limitations of their work, in the last part of the Discussion.

We believe this article is suitable for publication in the journal although some revisions are needed. The main strengths of this paper are that it addresses an interesting and very timely question and provides a clear answer, with some limitations.

We suggest a linguistic revision and the addition of some references for a matter of consistency. Moreover, the authors should better clarify some points.

Author Response

Thank you for your reviewing our manuscript. We have revised the manuscript in accordance with your valuable suggestions.

  • We cited the references and mentioned the current status of chemotherapy for gastric cancer in the Introduction. The prognosis for unresectable recurrent gastric cancer is extremely poor. Chemotherapy for gastric cancer is comprised mainly of cytotoxic agents, and the use of molecularly targeted agents is still rare. Only programmed cell death 1 (PD-1) and human epidermal growth factor receptor 2 have recently become new targets in advanced gastric cancer. In addition, the tumor microenvironment has emerged as a potential therapeutic target in several malignancies including gastric cancer. Several agents targeting the tumor microenvironment are currently under assessment in both preclinical and clinical studies. The development of chemotherapy based on the pathogenesis of gastric cancer is desired.
  • The revised manuscript was edited by a native speaker.
  • As you indicated, we included a discussion of future prospects for MC180295, which is expected to be a new epigenetic therapy target. A clinical study is underway to evaluate its safety. If MC180295 can be administered to EBV-associated gastric cancer, it may restore expression of tumor suppressor genes whose expression is suppressed by DNA methylation and histone modification. Tumor-associated antigens are also down-regulated in EBV-associated gastric cancer through epigenetic mechanisms, and thus restoration of their expression would be an attractive approach for possible anti-tumor immunity or for combination therapy with immune checkpoint inhibitors. We believe that this would be a major advance in the chemotherapy of unresectable EBV-associated gastric cancer.
  • The limitation of this study is that we only examined the growth-inhibitory effect of MC180295 in EBV-positive GC lines in vitro. We are currently planning to investigate the effect of MC180295 on cancer cell lines transplanted into immunodeficient mice.

Reviewer 2 Report

  1. In the cell cycle analysis, MC180295 increased the % of sub-G1 population and decreased the % of G2/M population. However, there is no effect on YCCEL1. More time points should be tested in order to confirm the effect of MC180295 on cell cycle.
  2. In line 86 and 87, the information of “Table 2” and “Table 3” are incorrect.
  3. Demethylating agents would generally increase gene expression. The authors showed that MC180295 decreased expression of several genes involve in DNA repair and cell cycle regulation. Would the reduction of gene expression merely due to cell death induced by the drug?
  4. The effect of the drug on the production of proteins involved in DNA repair and cell cycle should also be investigated.
  5. There are several papers showing that reactivation of EBV lytic cycle by histone deacetylase inhibitors (e.g. SAHA) could lead to enhanced cell death. In figure 4, the authors showed that MC180295 could also induce the expression of EBV lytic and latent proteins. Is there any causal relationship between the expression of EBV lytic and latent genes and human DNA repair genes?

Author Response

Thank you for your reviewing our manuscript. We have revised the manuscript in accordance with your valuable suggestions.

  1. We reexamined cell cycle analysis of MC180295-treated YCCEL1 cells and found that the percentage cells in S phase decreased and those in the sub-G1 fraction increased at 48 h and 72 h after the treatment. The graph of results was changed to a new one.
  2. We corrected the table numbers in section 2.2.
  3. In the Serial Analysis of Gene Expression (SAGE) experiment, we used mRNA from YCCEL1 and SNU719 cells treated with 1 μM MC180295 for 48h. As the reviewer speculated, we might evaluate the reduction of gene expression due to cell death by MC180295. But, Zhang et al. report mentions that MC180295 functions as CDK9 inhibitor and may suppress cell cycle progression. Moreover, other reports mention that MC180295 suppresses the expression of genes related to the DNA repair cascade because CDK9 is involved in the DNA repairment.
  4. Expression of proteins involved in DNA repair and cell cycle regulation were investigated. In SNU719 cells treated with MC180295, expression of γH2AX, a sensitive marker of DNA damage response, showed increase up to 24 hours, but showed slight decrease at 48 h. In YCCEL1 cells, MC180295 treatment increased γH2AX expression up to 48 h. Caspase3 plays important role in the process of apoptosis and detection of cleaved caspase3 indicates the occurrence of apoptosis. In SNU719 cells treated with MC180295, expression of cleaved caspase 3 showed increase up to 24 h and showed slight decrease at 48 h. In YCCEL1 cells, MC180295 treatment increased cleaved caspase 3 expression at 48 h. The expression of cleaved caspase3 and γH2AX increased in the similar fashion. The result showed that the data from the cell cycle analysis were convincing.
  5. BZLF1 has been shown to be a binding partner of several DNA damage response proteins, such as 53BP1, Ku80, p53 and RNF8. Moreover, screening of EBV proteins that inhibit the DNA damage response identified EBV BKRF4 as a DNA damage response inhibitor. Because BKRF4 was expressed in both latent and lytic EBV infections, several EBV genes expressed during both lytic and latent infection might be involved with DNA damage response.

Reviewer 3 Report

The present manuscript investigates the effect of a demethylating agent MC180295 on Epstein-Barr virus-associated gastric carcinoma (EBVaGC) cell lines. The authors proceed to evaluate the effect of this inhibitor on the expression of DNA repair genes and cell cycle as a potential explanation for the effect of the inhibitor on these cell lines. The study's premise is promising; however, the study would benefit from including the following points to make the conclusions convincing.

  1. The introduction is minimal and could benefit from more information about the rationale for focusing on the DNA repair pathways etc.
  2. Results: Figure 1: The authors have selected two EBVaGC cells and the MKN28 cell line with and without EBV to show the specific inhibition of cell lines with EBV. However, for the YCCEL1 and SNU719 cell lines, a dose-dependent effect of the MC180295 inhibitor on a control gastric carcinoma (without the EBV) is not provided. Could the authors potentially offer this comparison to show the specificity of the inhibition for cell lines with the EBV? While the MKN28 does give this evidence to a limited extent, a dose-dependent analysis is not available.
  3. Figure 1: For the MKN28 cell line, could the authors present a DMSO comparison and mention the inhibitor's dose used for the assay?
  4. Figure 1: The figure legend mentions that growth inhibition is significant after 48hrs. However, the authors have not provided the statistical value for this. Could the authors provide the statistical analyses used to make this conclusion?
  5. Figure 1: The Y-axis label could be changed to ‘absorbance at 490nm’.
  6. Results: Figure 2: Similar point as above, i.e., a non-EBV cell line could serve as a control for the assay to make the data more robust.
  7. The table numbers for section 2.2 do not match the description in the test
  8. “The expression of genes belonging to Role of BRCA1 in DNA Damage Response and Cell Cycle Control Chromosomal Replication’ is a confusing statement. Could the author clarify the statement?
  9. Could the authors provide a rationale for selecting the genes FANCM, BRCA2, RAD51, TOP2A, and CDC45? For the fold change in expression, could the authors give a graph with maybe a delta Ct method of calculation?
  10. Figure 3: Is the reduction statistically significant? Could the authors apply a statistical test to show the difference? Could the authors indicate the baseline expression of these genes in cell lines without EBV?
  11. Could the authors please include statistical analysis and the number of replicates/trials done for each experiment in the materials and methods?

Author Response

Thank you for your reviewing our manuscript. We have revised the manuscript according to your valuable suggestions.

  1. We mentioned the current status of chemotherapy for gastric cancer in the Introduction. Since the results of the SAGE analysis focused on the DNA repair pathway and the BRCA pathway, an explanation was added to the Results (From line 98 to line 108 in the clean version).
  2. YCCEL1 and SNU719 cell lines are 100% EBV-positive cell lines established from EBV-associated gastric cancer in vivo; thus, we were unable to compare their growth inhibitory effects with those of EBV-negative sublines.

We omitted a dose-dependent analysis for MKN28. MKN28 cells with or without EBV were treated with 1 µM MC180295, which showed a difference in cell proliferation from the control based on the results of dose-dependent analysis for YCCEL1 and SNU719.

  1. For the MKN28 cell line, we added data on the growth curve of MKN28 with or without EBV in DMSO-containing medium as a control in Figure 1(c). We used 1 μM MC180295 for the MKN28 cell line. We added this information to the Materials and Methods.
  2. The inhibition of cell growth of SNU719 by MC180295 was significant at 2 to 4 days after exposure to 1 μM MC180295, whereas a significant difference was found after 4 days in YCCEL1. We added the statistical value in Figure 1a–c and mentioned in the footnote of each figure. We explained the statistical analysis in the Materials and Methods (From line 270 to line 273 in the new version).
  3. We corrected the Y-axis label to “Absorbance at 490 nm”.
  4. We were unable to show data of the cell cycle analysis with EBV-negative gastric cancer lines. As we mentioned in the comment for second question, YCCEL1 and SNU719 cell lines are 100% EBV-positive cell lines established from EBV-associated gastric cancer in vivo; thus, we were unable to compare their growth inhibitory effects with those of EBV-negative sublines.
  5. We corrected the table numbers for section 2.2.
  6. We explained that “Role of BRCA1 in DNA damage response” and “Cell cycle control chromosomal replication” are the names of canonical pathways in Ingenuity Pathway Analysis. The results of SAGE were integrated by Ingenuity Pathways Analysis. We found two pathways in which gene expression varied significantly. The expression of genes belonging to “Role of BRCA1 in DNA damage response” and “Cell cycle control chromosomal replication” were significantly reduced in both YCCEL1 and SNU719 cells. We explained genes included in “Role of BRCA1 in DNA damage response” and “Cell cycle control chromosomal replication” on the Table 1 and Table 2. We believe now our statement becomes more clear.
  7. Among the genes belonging to the pathways, we selected genes that were particularly relevant in the literature (Ref. 15, 20-23) and performed quantitative PCR.
  8. We added statistical information to the caption of Figure 3. We could not indicate the expression level of these genes in cell lines, because there are no EBV-negative counterpart cells. As we mentioned in the comment for second question, YCCEL1 and SNU719 cell lines are 100% EBV-positive cell lines established from EBV-associated gastric cancer in vivo; thus, we were unable to examine the expression level of these genes with those of EBV-negative sublines.
  9. We explained the statistical analysis in the Materials and Methods (From line 270 to line 273 in the new version). We mentioned that data were obtained from three independent experiments in in the Materials and Methods and the footnote of each figure.

Round 2

Reviewer 1 Report

acceptance.

Reviewer 2 Report

The overall quality of the manuscript is improved.